# Simultaneous Detection of Three Subgroups of Avian Leukosis Virus Using the Nanoparticle-Assisted PCR Assay

**DOI:** 10.3390/v16010015

**Published:** 2023-12-21

**Authors:** Miaoli Wu, Shuaiqi Hu, Yujun Zhu, Feng Cong, Shengwang Liu

**Affiliations:** 1Division of Avian Infectious Diseases, State Key Laboratory of Veterinary Biotechnology, Harbin Veterinary Research Institute, Chinese Academy of Agricultural Sciences, Harbin 150026, China; wml@gdlami.com; 2Guangdong Laboratory Animals Monitoring Institute and Guangdong Provincial Key Laboratory of Laboratory Animals, Guangzhou 510633, China; zhuyujun@gdlami.com; 3College of Animal Science, Anhui Science and Technology University and Anhui Province Key Laboratory of Animal Nutritional Regulation and Health, Fengyang 233100, China; hsqmcga@163.com

**Keywords:** nanoPCR, avian leukosis virus, discrimination

## Abstract

Nanoparticle-assisted polymerase chain reaction (nanoPCR) is a novel method for the rapid detection of pathogens. A sensitive and specific multiple nanoPCR assay was developed for simultaneous detection of avian leucosis virus (ALV) subgroups A, B and J. In this study, three pairs of primers were designed, based on the conserved region of the gp85 gene. An exploration of the optimal primer concentration and annealing temperature were carried out, for better performance of the nanoPCR assay. According to the results, the multiple nanoPCR assay amplified 336 pb, 625 bp and 167 bp fragments of ALV-A, -B and -J, respectively, and showed no cross-reactivity with irrelevant pathogens, suggesting the excellent specificity of the assay. The constructed standard DNA templates were used to estimate the limit of detection. As shown by the results, the detection limit of the nanoPCR assay was nearly 10 copies/μL. To further evaluate the detection ability of the assay, 186 clinical samples were detected using the nanoPCR assay, among which, 14 samples were confirmed as ALV positive; the results were further confirmed by sequencing. In conclusion, a highly specific and sensitive nanoPCR assay was successfully developed, which could be a useful tool for clinical diagnosis as well as for the discrimination of ALV-A, -B and -J.

## 1. Introduction

Avian leucosis, induced by avian leucosis virus (ALV), causes immunosuppression, growth deceleration and even fatal tumors in chickens [1,2,3]. In the past, ALV was prevalent in breeding flocks in China, since the virus could be transmitted either genetically or acquired via infection, causing significant economic losses to the poultry farming industry [4,5]. Additionally, evidence showed that ALV had spread not only in commercial chickens but also in backyard chickens; it is quite difficult to control the disease with such a large number of hosts [6,7]. In fact, as no effective vaccines were available, the main control strategy relied on the eradication project initiated by the local government [8]. Therefore, the strict and complete monitoring of ALV was essential to the poultry industry.

Avian leukosis virus (ALV) belongs to the *Alpharetrovirus* genus of the *Retroviridae* family, and it can be classified as either endogenous or exogenous virus, depending on its characteristics and transmission routes [9]. Exogenous ALVs can be further divided into 11 different subgroups, based on their host range, viral envelope interference and cross-neutralization mode. The subgroups A, B, C, D, J and K are specific to chicken, among which the A, B and J subgroups are common in the field, while subgroups C and D are rarely encountered [10]. Being the most epidemic strains, subgroups A, Band J have caused huge economic losses for poultry farms. Therefore, the detection and discrimination of subgroup A, B and J are crucial to the control and eradication of ALV.

Conventional methods for ALV identification, such as virus isolation, are time-consuming and labor-intensive. Furthermore, an additional follow-up detection is needed for final identification, when virus isolation is performed. With the development of molecular technology, PCR, quantitative real-time PCR and recombinase polymerase amplification (RPA) have been developed for the rapid detection of ALV in chickens [11,12,13,14]. Despite the obvious advantages, such as being time-saving, highly specific and sensitive, these methods have their own shortcomings. For example, expensive equipment and a well-trained operator are needed to perform real-time PCR assay. The basal RPA assay does not rely on a special device, but its low detection limit restricts its applications. As for the upgraded version, the real-time RPA, with the help of fluorescence labeling probes, has significantly improved its sensitivity, as well as its detection fee. However, most owners are not willing to pay for the bill, making the wide application of real-time RPA in poultry farms impossible [15]. Therefore, a rapid, sensitive and cost-effective detection method is in demand for ALV diagnosis in the field.

With the explosive revolution of nanoscience, nanotechnology-based approaches have permeated into many different industries, including food, chemicals, medicine and agriculture. Great efforts have been made to utilize nanomaterial (NMs) in the development of detection techniques [16]. Nanoparticle-assisted polymerase chain reaction (nanoPCR) is an advanced modification of PCR; the thermal conductivity is significantly enhanced with the help of solid gold nanometer particles [17,18]. The whole amplification process has been accelerated, following the addition of nanoparticles to the reaction system. The reduction in total reaction time has finally enhanced the efficiency of the PCR assay. Additionally, the special nanomaterial, such as magnetic NPs, polymer-modified silica, CNTs, etc., might form a competitive relationship with the DNA templates, which would help to improve the specificity of the PCR assay [19,20]. As a result, the performance of the PCR assay is significantly improved when assisted using nano-particles [21]. In the past few years, this technique has been widely used in pathogen identification in veterinary research; for example, a nanoparticle-assisted RT-PCR assay has been established as being able to distinguish porcine epidemic diarrhea virus (PEDV), bovine respiratory syncytial virus and canine coronaviruses [22,23,24].

In this study, a nanoparticle-assisted PCR assay was developed for the rapid detection of ALV. Three primer pairs were designed for the simultaneous detection and discrimination of subgroups A, B, and J of ALVs. Optimal reaction conditions were explored to facilitate a better performance of the newly developed nanoPCR assay. This novel assay could be a useful tool for the detection and differentiation of ALV subgroups, which would be really helpful in ALV eradication.

## 2. Materials and Methods

### 2.1. Viruses and Sample Collection

ALV-A (GD08 strain), was kindly offered by Dr. Cao Weisheng, and ALV-B (SDAU09E3 strain) and ALV-J (NX0101 strain) were kindly offered by Dr. Cui Zhizhong. Newcastle disease virus (NDV), avian infectious laryngotracheitis virus (ILTV), Marek’s disease virus (MDV), infectious bronchitis vaccine (IBV), infectious bursal disease virus (IBDV), reticuloendotheliosis virus (REV), ALV-C, ALV-E, ALV-K and avian encephalomyelitis virus (AEV) were preserved in our lab. A total of 120 clinical specimens, including the kidneys, livers and spleens of 6- to 7-week-old sick chickens, were collected from the elimination group of two different yellow feather broiler breeders in Huizhou, Guangdong province. Under the ALV eradication program, 66 anticoagulation blood samples were collected from two poultry farms in Jiangmen, Guangdong province. Each of the 186 samples were preserved at −80 °C before further uses.

### 2.2. Primer Design

Primers were designed using the Oligo 6.0 software (Molecular Biology Insights, Inc., Colorado Springs, CO, USA), following the manufacturer’s instructions [25]. There are three group-specific (gs) antigens of ALV, encoded by the gag, pol and env genes, respectively. The env gene encodes two proteins, namely gp85 and gp37; gp85 determines the ALV subgroup specificity, making it an ideal target for the discrimination of ALV subgroups [26]. Specific primers against ALV-A, ALV-B and ALV-J were designed to focus on the conserved region of gp85 gene. To differentiate the three subgroups of ALV, the amplicons should be clearly distinguished by the follow-up analysis; therefore, different sizes of the PCR products were amplified using different primer pairs (Table 1). The selected primer pairs were validated using online BLAST (Nucleotide BLAST: Search nucleotide databases using a nucleotide query (www.nih.gov)) to guarantee the accuracy of the PCR products. All the primer pairs were synthetized and purified by Sangong Biotech (Guangzhou, China).

### 2.3. Standard DNA Template Construction

Total RNA was directly extracted from DF-1 cells, 5 days after infection with ALV-A, ALV-B or ALV-J, using the automatic nucleic acid extraction instrument (Tiangen Biotech, Beijing, China) according to the manufacturer’s instruction. Reverse transcription was carried out, to acquire the cDNA of ALVs. Then, specific target fragments were amplified using the corresponding primers and separately cloned into the pMD-18T vector (Takara Biotechnology Co., Ltd., Dalian, China) following the manufacturer’s instructions. The recombinant plasmids clones (designated 18T-A, 18T-B and 18T-J, respectively) were confirmed by PCR assay and sequencing (Sangong Biotech, Guangzhou, China) before being further extracted with the Plasmid Mini Kit (Omega Bio-Tek, Norcross, GA, USA). The plasmid concentrations were determined by the Nanodrop 2000 (Thermo, Waltham, MA, USA) and the copy numbers were calculated based on the corresponding molecular weight and plasmid concentration. The standard DNA templates were stored at −20 °C before they were subjected to a conventional multiple PCR assay and the novel multiple nanoPCR assay as described in the following sections.

### 2.4. Establishment of the Conventional Multiple PCR Assay

The conventional multiple PCR assay was performed for comparison with the multiple NanoPCR. A total 20 μL volume containing 1 μL of each viral cDNA; 1 μL of each primers; 2 μL of dNTPs; 1 μL of KOD FX Neo (1 U/μL) (TOYOBO Biotechnology Company, Shanghai, China); 5 μL of 2× PCR buffer for KOD FX Neo (TOYOBO Biotechnology Company, Shanghai, China). The PCR reaction conditions were as follows: initial denaturation at 94 °C for 5 min, followed by 30 cycles of denaturation at 94 °C for 40 s, annealing at 58 °C for 40 s, and an extension at 72 °C for 60 s, with a final extension at 72 °C lasting for 10 min. The amplified products were analyzed by electrophoresis on 2% agarose gels.

### 2.5. Establishment of the Multiple NanoPCR Assay

#### 2.5.1. Optimization of the Multiple NanoPCR Assay

Experiments were carried out to optimize the annealing temperature, primer concentration and template volume to improve the performance of the nanoPCR assay. Briefly, the optimal annealing temperature was determined using the primer concentrations and template concentrations indicated above. Once the optimal annealing temperature was determined, optimal primer concentrations and nucleic acid templates were further determined, based on the optimal annealing temperature. Specifically, the cDNA templates were tested, ranging from 0.2 μL to 1.4 μL, while the primers at a 10 μM working concentration were also tested, ranging from 0.2 μL to 1.2 μL per reaction volume. The reaction volume also contained 0.5 μL of *Taq* DNA polymerase (5 U/μL) (GRED, Weihai, China) and 10 μL of 2× nanoPCR Buffer (GRED, Shandong, China), with ddH_2_O up to 20 μL. The PCR reaction conditions were as follows: 94 °C for 5 min, followed by 30 cycles of 94 °C for 40 s, annealing ranging from 52 °C to 61 °C for 30 s, and 72 °C for 60 s, with a final extension at 72 °C for 10 min. The amplified products were analyzed by 2% agarose gel electrophoresis, as usual.

#### 2.5.2. Sensitivity Analysis of the Multiple NanoPCR Assay

To determine the detection limit of the multiple nanoPCR assay, tenfold serial dilutions of standard plasmid DNA, ranging from 6.31 × 10^9^ copies/μL to 6.0 × 10^0^ copies/μL of 18T-A, 3.51 × 10^9^ copies/μL to 4 × 10^0^ copies/μL of 18T-B and 3.32 × 10^9^ copies/μL to 3.0 × 10^0^ copies/μL of 18T-J, were tested using the multiple nanoPCR assay. In the meantime, a parallel experiment on the conventional multiple PCR assay was carried out for sensitivity comparison. All the samples were tested in triplicate and the amplified products were analyzed by electrophoresis, as above.

#### 2.5.3. Specificity of the Multiple NanoPCR Assay

A specificity analysis was carried out, to avoid false positives with other common pathogens in chickens. The DNA or cDNA of the other viruses, including Newcastle disease virus (NDV), avian infectious laryngotracheitis virus (ILTV), Marek’s disease virus (MDV), infectious bronchitis vaccine (IBV), infectious bursal disease virus (IBDV), avian encephalomyelitis virus (AEV), reticuloendotheliosis virus (REV) and the ALV subgroups C, E and K, were separately subjected to the multiple nanoPCR and the conventional PCR assays, for specificity analysis. Meanwhile, the plasmid DNA of ALV-A, ALV-B and ALV-J were used as a positive control. All the PCR products were analyzed by 2% agarose gel.

### 2.6. Evaluation of the Multiple NanoPCR Assay with Clinical Specimens

A total of 186 clinical samples were collected from four different poultry farms in Guangdong province. Among them, 120 samples belonged to sick chickens showing typical clinical symptoms, including gray-white nodules in the liver or kidney and abnormal enlargement of the liver and spleen. The rest of the samples were from healthy chickens that had been through the ALV eradication project. The reverse transcription cDNA of all these samples were subjected to the newly developed nanoPCR assay. All the samples were tested in duplicate. In addition, all the PCR amplicons were then cloned into pMD-18T vectors, and the positive clones were sent for sequencing by Sangong Biotech (Guangzhou, China). The sequencing results were further analyzed using the online software BLASTN (www.ncbi.nlm.nih.gov/blast, accessed on 10 December 2021).

## 3. Results

### 3.1. Optimization of the Multiple NanoPCR Assay

Reverse transcription cDNA was used as a template during the optimization of the nanoPCR assay. The screening annealing temperatures ranged from 52 to 61 °C. As shown by the results, the higher the annealing temperature, the more acquiring amplified products. This tendency was more obvious during ALV-B amplification than for the other two subgroups. Not many differences were found between annealing at 58 °C, 59 °C, 60 °C and 61 °C. Therefore, 58 °C was chosen as the optimal annealing temperature for the following experiments (Figure 1).

Once the annealing temperature was determined, the optimal primer concentrations were further assessed. Six different concentrations were tested, as shown in Figure 2, but not much difference was found between the different concentrations of each primer (each different concentration encompassed a slight increase of 0.2 μL) during the PCR assay. However, the PCR assay with the highest primer concentration produced more amplicons than the lowest one (Figure 2). Since no visible differences were found between the concentrations of 0.8 μL, 1.0 μL and 1.2 μL, 0.8 μL was chosen as the optimal primer concentration, for cost-saving purposes.

When the optimal annealing temperature (58 °C) and primer concentration (0.8 μL) were determined, the optimal template volume was confirmed. According to the results, the three subgroups of ALV shared different optimal templates, as shown in Figure 3: 0.8 μL of ALV-B, and 1 μL of ALV-A and ALV-J presented the best amplified results (Figure 3).

All the subsequent experiments were carried out based on these optimized conditions; briefly, the multiple nanoPCR assay was performed using a 20 μL reaction mixture that included: 0.8 μL of the ALV-B and 1.0 μL of the ALV-A and ALV-J reverse transcription cDNA; 10 μL of 2× Nano-buffer; 0.8 μL of each primer pair (working concentration was at 10 μM); 0.5 μL of *Taq* DNA polymerase (5 U/μL) and ddH_2_O up to 20 μL. The PCR reaction conditions were as follows: 94 °C for 5 min, 30 cycles of 94 °C for 40 s, 58 °C for 40 s, and 72 °C for 60 s, with a final extension at 72 °C for 10 min.

### 3.2. Sensitivity of the Multiple NanoPCR Assay

The sensitivity of the multiple nanoPCR assay and the conventional multiple PCR assay were compared by using serial dilutions of the 18T-A, 18T-B and 18T-J as templates. For ALV-A, the detection limit was 6.31 × 10^3^ copies/μL for the conventional multiple PCR assay but 6.31 × 10^1^ copies/μL for the multiple nanoPCR assay. The detection limit of ALV-B was 3.51 × 10^3^ copies/μL for the conventional multiple PCR assay but 3.51 × 10^1^ copies/μL for the multiple nanoPCR assay. As for ALV-J, the detection limit was 3.32 × 10^4^ copies/μL for the conventional multiple PCR assay but 3.32 × 10^1^ copies/μL for the multiple nanoPCR assay (Figure 4). These results indicated that the multiple nanoPCR assay was 100 times more sensitive than the conventional multiple PCR assay.

### 3.3. Specificity of the Multiple NanoPCR Assay

Neither the multiple nanoPCR assay nor the conventional multiple PCR assay would amplify the fragments of ALV without the relative target cDNA serving as a template (Figure 5). As shown by the results, no cross-reactions with NDV, ILTV, MDV, IBV, IBDV AEV, REV, ALV-C, ALV-E or ALV-K were found during the amplification, suggesting the high specificity of the nanoPCR assay.

### 3.4. Evaluation of the NanoPCR Assay Using Clinical Specimens

A total of 186 clinical specimens were tested using the nanoPCR assay. Independent sequencing was performed to confirm the results of the nanoPCR assay. As shown in Table 2, fourteen specimens tested positive for ALV infection, among which eleven specimens were ALV-J positive, one specimen displayed a co-infection with ALV-A and ALV-J, and two specimens displayed co-infections with ALV-B and ALV-J. Independent sequencing confirmed that the test results of the nanoPCR assay showed 100% coincidence with sequencing, indicating the accuracy of the nanoPCR assay and its potential application for clinical ALV diagnosis.

## 4. Discussion

Avian leucosis (AL), caused by avian leucosis virus, has brought huge economic losses to the poultry breeding industry in China [27]. Since no effective vaccine is available for ALV control, detection and differential diagnosis are essential for ALV monitoring. In the past decades, the increasing occurrence of ALV in chicken farms has led to a demand for the rapid and accurate identification of ALV in clinical samples. Traditional methods, such as virus isolation, had previously been used for ALV identification and served as the “gold standard”. However, it took about 7 days to present the results. Additionally, because the infection of ALV in DF-1 cells showed no typical cytopathic effect (CPE), an extra test was needed for a final conclusion. Usually, the cell cultures were subjected to a commercial enzyme-linked immunosorbent assay (ELISA) kit for the analysis of ALV proliferation in DF-1 cells. To make it worse, the detection ability of the available commercial avian leucosis virus antigen test kits was unstable, leading to false negative results, which hampered the eradication of ALV.

Some molecular biology techniques have been applied to ALV diagnosis; for example, conventional PCR. As we know, a complete conventional PCR procedure can last for about two hours, accompanied by half an hour of agarose gel electrophoresis. Though its time-consuming and labor-intensive characteristics are acceptable, its low sensitivity limits the application of conventional PCR methods. To conquer this sensitivity limitation, a real-time PCR assay was developed for ALV detection. With the help of fluorescence-modified probes, the detection limit of ALV reached as low as 10 viral DNA copies [28]. However, much more expensive instruments, as well as well-trained operators, were needed to perform real-time PCR, both of which were not available on most of the poultry farms. Therefore, it was urgent to develop a sensitive and cost-effective method to meet the needs of clinical sample detection on poultry farms.

Nanomaterials are widely used as helpers during PCR amplification; thanks to the excellent thermal conductivity of gold nanoparticles, the non-target temperature has been greatly reduced during the amplification process [29]. As a result, the addition of solid gold nanometal particles in a nanoPCR assay triggered a remarkable increase in specificity and sensitivity performances, when compared with conventional PCR methods [30,31]. Therefore, it could serve as a perfect substitution for conventional PCR for ALV detection in clinical samples.

In the present study, a multiple nanoPCR assay was developed. The subgrouping determinant gene gp85 was used as the target for ALV identification [32]. Three primer pairs focused on the conserved region of the gp85 gene, which were designed from ALV-A, ALV-B and ALV-J. The lengths of the amplicons were displayed at different sizes, so were distinguished using agarose gel electrophoresis. In order to obtain better results with the multiple nanoPCR assay, optimal reaction conditions, including annealing temperature, primer concentration and template concentration, were explored. As shown by the agarose gel electrophoresis, differences were mainly found in the longest amplicon, referring to ALV-B. Unlike ALV-B, during the optimal reaction condition screening, ALV-A and ALV-J displayed no obvious differences under different conditions. Just like the markers, the lower bands tended to become blurred; the 336 bp of the ALV-A amplicon and the 167 bp of the ALV-J amplicon were not clear enough to tell the difference. However, it was impossible to run three different agarose gel electrophoresis analyses for each subgroup, when amplicons were produced by the multiple PCR reaction. Considering that, the nanoPCR assay was developed for the simultaneous detection of three subgroups of ALVs, so the optimal reaction conditions for each subgroup were pointless. In this case, the optimal condition of the whole multiple nanoPCR assay was mostly determined by ALV-B. In this study, not only the annealing temperature but also the primer and template concentration were assessed; however, the optimal template concentration was not as meaningful as the annealing temperature. Usually, the PCR assay is carried out following the manufacturer’s instructions, which would recommend a volume of DNA/RNA template. However, the precise concentration of the DNA/RNA template would not be clearly confirmed before being subjected to the PCR assay.

Despite the same primer pairs being used, the detection limit of the conventional PCR assay was much higher than the nanoPCR assay. According to the results, the detection limit of the nanoPCR was about 10^1^ copies/μL for ALV-A, ALV-B and ALV-J, while the conventional PCR methods could only detect 10^3^ copies/μL of the relative DNA standards. With such an excellent detection ability, the nanoPCR assay could be a strong competitor for the real-time PCR assay, since it does not rely on expensive equipment. Though both the nanoPCR assay and the real-time PCR assay shared a similar detection limit, the operation of the nanoPCR assay was less convenient than the real-time PCR assay, as real-time monitoring could not be achieved, and the PCR products needed to be analyzed by agarose gel electrophoresis analysis. Considering that the cost of the nanoPCR assay is much cheaper and the real-time PCR equipment is not available in most of poultry farms, it is believed that the nanoPCR assay would be a better option for ALV detection on poultry farms.

In this study, a total of 186 clinical specimens were collected and tested using the nanoPCR assay. The results showed that fourteen specimens were infected with ALV-A, ALV-B or ALV-J; eleven specimens displayed a single infection while three specimens displayed co-infections. All the detection results were further confirmed by sequencing, to ensure the accuracy of the nanoPCR assay. As revealed by the results, the newly developed nanoPCR assay was capable of detecting ALV correctly, making it an ideal tool for ALV-A/B/J detection in clinical samples.

## 5. Conclusions

In conclusion, a rapid, sensitive and specific multiple nanoPCR assay was developed. This novel assay could simultaneously detect ALV-A, ALV-B and ALV-J and could be applied to ALV diagnosis for epidemiological and pathological study, as well as infectious disease control.

## Figures and Tables

**Figure 1 viruses-16-00015-f001:**
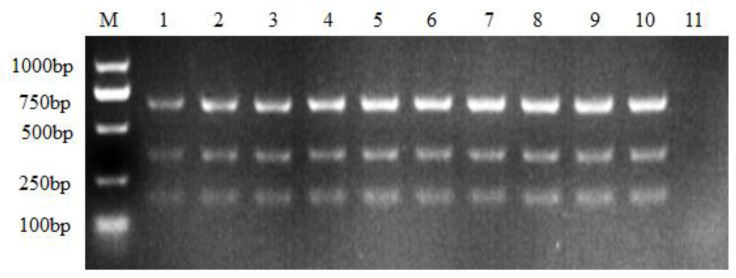
Optimization of annealing temperature. M: DL1000 DNA marker. The annealing temperature ranges from 52 °C to 61 °C. 1: 52 °C; 2: 53 °C; 3: 54 °C; 4: 55 °C; 5: 56 °C; 6: 57 °C; 7: 58 °C; 8: 59 °C; 9: 60 °C; 10: 61 °C; 11: negative control with no ALV nucleic acids.

**Figure 2 viruses-16-00015-f002:**
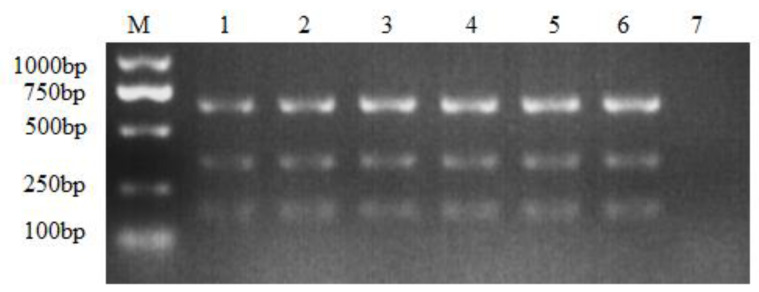
Optimization of the primer concentrations. M: DL1000 DNA Marker; the primer volume ranged from 0.4 μL to 1.4 μL of each primer under the working concentration of 10 μM. 1: 0.4 μL; 2: 0.6 μL; 3: 0.8 μL; 4: 1.0 μL; 5: 1.2 μL; 6: 1.4 μL; 7: negative control.

**Figure 3 viruses-16-00015-f003:**
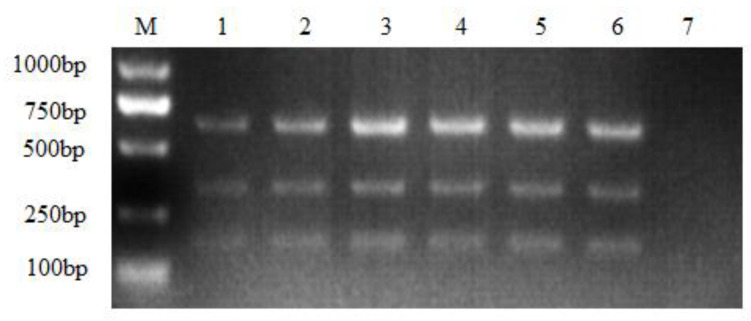
Optimization of the template concentration. M: DL1000 DNA Marker; the cDNA volume ranged from 0.2 μL to 1.2 μL. 1: 0.2 μL; 2: 0.4 μL; 3: 0.6 μL; 4: 0.8 μL; 5: 1.0 μL; 6: 1.2 μL; 7: negative control.

**Figure 4 viruses-16-00015-f004:**
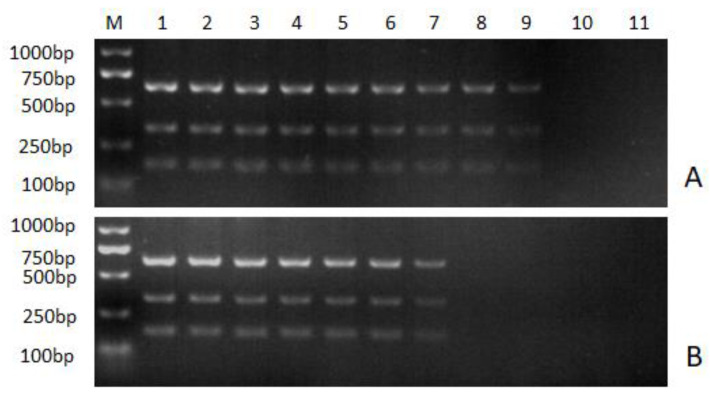
Sensitivity analysis of the multiple NanoPCR assay. (**A**): nanoPCR assay; (**B**) conventional multiple PCR assay. M: DL1000 marker; 1–10: tenfold serial dilutions of standard DNA plasmids. Namely, 1: 6.31 × 10^9^ copies/μL for ALV-A, 3.51 × 10^9^ copies/μL for ALV-B and 3.32 × 10^9^ copies/μL for ALV-J; 2: 6.31 × 10^8^ copies/μL for ALV-A, 3.51 × 10^8^ copies/μL for ALV-B and 3.32 × 10^8^ copies/μL for ALV-J; 3: 6.31 × 10^7^ copies/μL for ALV-A, 3.51 × 10^7^ copies/μL for ALV-B and 3.32 × 10^7^ copies/μL for ALV-J; 4: 6.31 × 10^6^ copies/μL for ALV-A, 3.51 × 10^6^ copies/μL for ALV-B and 3.32 × 10^6^ copies/μL for ALV-J; 5: 6.31 × 10^5^ copies/μL for ALV-A, 3.51 × 10^5^ copies/μL for ALV-B and 3.32 × 10^5^ copies/μL for ALV-J; 6: 6.31 × 10^4^ copies/μL for ALV-A, 3.51 × 10^4^ copies/μL for ALV-B and 3.32 × 10^4^ copies/μL for ALV-J; 7: 6.31 × 10^3^ copies/μL for ALV-A, 3.51 × 10^3^ copies/μL for ALV-B and 3.32 × 10^3^ copies/μL for ALV-J; 8: 6.31 × 10^2^ copies/μL for ALV-A, 3.51 × 10^2^ copies/μL for ALV-B and 3.32 × 10^2^ copies/μL for ALV-J; 9: 6.31 × 10^1^ copies/μL for ALV-A, 3.51 × 10^1^ copies/μL for ALV-B and 3.32 × 10^1^ copies/μL for ALV-J; 10: 6.31 × 10^0^ copies/μL for ALV-A, 3.51 × 10^0^ copies/μL for ALV-B and 3.32 × 10^0^ copies/μL for ALV-J; 11: negative control.

**Figure 5 viruses-16-00015-f005:**
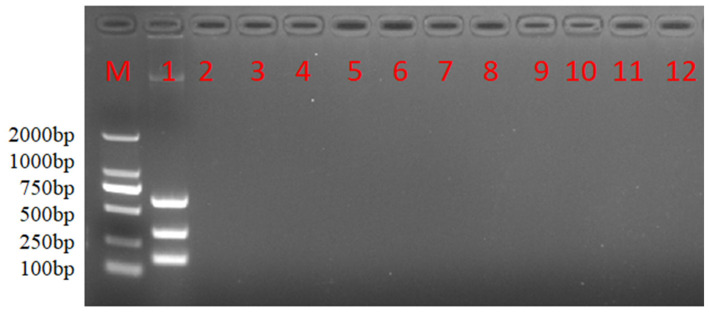
Specificity analysis of nanoPCR assay. M: DL2000 DNA Marker; 1~6: nucleic acids of different viruses. Namely, 1: ALV-A, ALV-B and ALV-J; 2: NDV; 3: ILTV; 4: MDV; 5: IBV; 6: IBDV; 7: AEV; 8: REV; 9: ALV-C; 10: ALV-E; 11: ALV-K; 12: negative control.

**Table 1 viruses-16-00015-t001:** Details of primers for ALV.

Name	Sequence	Target	Sequence ID	Site	Amplicon
AP1	ACTGGCGGCCCTGACAACAG	ALV-A	HM775328.1	219–555	336 bp
AP2	CGCACCGCAATACTCACTCCC				
BP1	CTACAACTGTTGGGTTCCCAGT	ALV-B	JF826241.1	482–1107	625 bp
BP2	GACCCCCTACCGGACGACTGG				
JP1	ACAAGCAAGAAAGACCCGG	ALV-J	DQ115805.1	82–248	167 bp
JP2	GTCATATTCGCCCAGGTGA				
AP1	ACTGGCGGCCCTGACAACAGCA	ALV-A	HM775328.1	219–555	336 bp
AP2	CGCACCGCAATACTCACTCCC				
BP1	CTACAACTGTTGGGTTCCCAGTCTCT	ALV-B	JF826241.1	482–1107	625 bp
BP2	GACCCCCTACCGGACGACTGGG				
JP1	ACAAGCAAGAAAGACCCGG	ALV-J	DQ115805.1	82–248	167 bp
JP2	GTCATATTCGCCCAGGTGA				

**Table 2 viruses-16-00015-t002:** The results of the nanoPCR assay for clinical specimens detection.

Virus	Nano-PCR Positive Specimens	Sequencing Positive Specimens	Coincidence Rate (%)
ALV-A	0	0	100
ALV-B	0	0	100
ALV-J	11	11	100
ALV-A + B	0	0	100
ALV-A + J	1	1	100
ALV-B + J	2	2	100

## Data Availability

The data presented in this study are available on request from the corresponding author.

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
