# Peer review of "Simultaneous Detection of Three Subgroups of Avian Leukosis Virus Using the Nanoparticle-Assisted PCR Assay"

_viruses, 2023, doi:10.3390/v16010015_

Round 1

Reviewer 1 Report

Comments and Suggestions for Authors

This paper notes that establishment of a multiplex nanoPCR system for ALVs detection that is more sensitive than conventional PCR and less expensive than real-time PCR. 

I think this report is interesting, but I have some concerns:

1) I think that one of the features of your PCR system is the use of 'normal' Taq polymerase. Why did you use 'normal' Taq? What would happen if a high efficiency enzyme was used? And how economical is nanoPCR buffer compared to using high efficiency PCR enzymes?

2) Although ALV-C and D are rare, will you intend to detect those two subgroups in your PCR system to make it better?

And you have to correct the following points: 

1) Please describe the amplicon-sequencing method.

2) Please indicate the primer design sites and the accession number of the reference sequences in Table 1.

3) Please add the legend for Lane 9 in Figure 4.

Author Response

Reviewer 1

  • I think that one of the features of your PCR system is the use of 'normal' Taq polymerase. Why did you use 'normal' Taq? What would happen if a high efficiency enzyme was used? And how economical is nanoPCR buffer compared to using high efficiency PCR enzymes?

A: In this study, high efficiency enzyme was used to develop the conventional multiple PCR assay, while “normal” Taq DNA polymerase were used for nanoPCR assay development. The nanoPCR kits we purchased also offered the Taq DNA polymerase. They claimed that the DNA polymerase they offered was high efficiency. Considering that, we didn’t need to explore the concentration of polymerase, we thought it might be better to use the polymerase recommended by the manufacturer.  So that we didn’t realize the option of replacing the Taq DNA polymerase with other high efficiency PCR enzymes. Actually, the nanoPCR kit was a little expensive than the high efficiency PCR enzymes, however, the increase of detection fee for each sample is negligible. Owing to its excellent performance in PCR amplification, nanoPCR possess much higher sensitivity than conventional PCR, making it a better option in clinical sample detection.

  • Although ALV-C and D are rare, will you intend to detect those two subgroups in your PCR system to make it better?

A: At the beginning, we had tried to identify 6 different subgroups of ALVs namely A, B, C, D, J and K. Thought, we use the agarose gel electrophoresis for PCR products analysis, it’s possible to identify 6 different sizes of brands when using the proper DNA marker. However, the primers would interfere each other during multiple PCR amplification and we had tried different primers, yet the results were not satisfying. Considering that, ALV-A, B, J were the most common subgroups we had encountered during clinical sample detection, so we just developed the nanoPCR for these three subgroups. Yes, we will try to develop a detection method that can cover as more subgroups as possible, more works should be done and we are still working on it.

And you have to correct the following points: 

  • Please describe the amplicon-sequencing method.

A: Usually, the amplicon can be sent out to the sequencing company for sequencing directly or they should be cloned into the vectors before further sequencing. In this study, all the amplicons were cloned into the pMD-18T vector and the positive clones were sent out for sequencing. We have to admit that we are not so familiar about the whole process of sequencing, usually we just received the sequencing results from the company. Yet proper changes had been made in the revised manuscript to emphasize that amplicons were cloned into vectors before sequencing.

  • Please indicate the primer design sites and the accession number of the reference sequences in Table 1.

A: Thank you for your kind advice. Changes had been made in Table 1,

  • Please add the legend for Lane 9 in Figure 4.

 A: We are really sorry for the careless. Changes had been made in Figure 4.

Reviewer 2 Report

Comments and Suggestions for Authors

In this study, Wu et.al have reported Nanoparticle-assisted polymerase chain reaction developed for detection of ALV-A, -B and –J. This study is very interesting. However, there are some limitations.

1  The abstract is a little too much, and some background are not needed. Please modified the abstract.

2  The missing of space was revealed all over the paper. Please check and correct them one by one.

The most major limitations I concerned is that Avian leukosis virus Subgroup E, D and K, and Reticuloendotheliosis Virus should be detected for specificity tests of the multiple nanoPCR assay.

Overall, the flow of this paper and grammar should be carefully checked when submission of revision version.

Comments on the Quality of English Language

Overall, the flow of this paper and grammar should be carefully checked when submission of revision version.

Author Response

Dear reviewer,

Thank you very much for your kind advice.

We are really sorry for the language mistakes we had made in the manuscript. We had carefully checked and collected them.

The abstract had been modified in the revised version.

The new specificity analysis had been performed. ALV-C, ALV-E, ALV-K and Reticuloendotheliosis virus (REV) had been detected and no cross-reaction was found. We have to confess that we failed to obtain ALV-D in time to carried out the experiment. However, sequence alignment had been performed to ensure no cross-reaction between ALV-D and the other three subgroups. The primers we used in this study could not match any sequences of ALV-D, we believe that cross-reaction would not happen.

Round 2

Reviewer 2 Report

Comments and Suggestions for Authors

I am satisfied with the revised version.